# Lower Respiratory Tract Infection and Genus Enterovirus in Children Requiring Intensive Care: Clinical Manifestations and Impact of Viral Co-Infections

**DOI:** 10.3390/v13102059

**Published:** 2021-10-14

**Authors:** Daniel Penela-Sánchez, Jon González-de-Audicana, Georgina Armero, Desiree Henares, Cristina Esteva, Mariona-Fernández de-Sevilla, Silvia Ricart, Iolanda Jordan, Pedro Brotons, María Cabrerizo, Carmen Muñoz-Almagro, Cristian Launes

**Affiliations:** 1Paediatrics Department, Hospital Sant Joan de Déu, 08195 Barcelona, Spain; daniel.penela@sjd.es (D.P.-S.); georgina.armero@sjd.es (G.A.); mariona.fernandez@sjd.es (M.-F.d.-S.); silvia.ricart@sjd.es (S.R.); 2Paediatrics Intensive Care Unit, Hospital Sant Joan de Déu, 08195 Barcelona, Spain; yolanda.jordan@sjd.es; 3Enterovirus and Viral Gastroenteritis Unit, Centro Nacional de Microbiología, Instituto Carlos III, 28222 Madrid, Spain; jonma_audikana@hotmail.com (J.G.-d.-A.); mcabrerizo@isciii.es (M.C.); 4Grupo de Investigación en Enfermedades Infecciosas Pediátricas, Institut de Recerca Sant Joan de Déu, Hospital Sant Joan de Déu, 08195 Barcelona, Spain; desiree.henares@sjd.es (D.H.); cristina.esteva@sjd.es (C.E.); pedro.brotons@sjd.es (P.B.); carmen.munoza@sjd.es (C.M.-A.); 5Molecular Microbiology Department, Hospital Sant Joan de Déu, 08195 Barcelona, Spain; 6Centro de Investigación Biomédica en Red de Epidemiología y Salud Pública (CIBERESP), Instituto de Salud Carlos III, 28029 Madrid, Spain; 7Paediatrics Department, Faculty of Medicine and Health Sciences, Universitat de Barcelona, 08007 Barcelona, Spain; 8Department of Medicine, School of Medicine, Universitat Internacional de Catalunya, 08017 Barcelona, Spain

**Keywords:** rhinovirus, enterovirus, co-infection, intensive care units, child, lower respiratory tract infection

## Abstract

Infection by rhinovirus (RV) and enterovirus (EV) in children ranges from asymptomatic infection to severe lower respiratory tract infection (LRTI). This cohort study evaluates the clinical impact of RV/EV species, alone or in codetection with other viruses, in young children with severe LRTI. Seventy-one patients aged less than 5 years and admitted to the Paediatric Intensive Care Unit (PICU) of a reference children’s hospital with RV or EV (RV/EV) LRTI were prospectively included from 1/2018 to 3/2020. A commercial PCR assay for multiple respiratory pathogens was performed in respiratory specimens. In 22/71, RV/EV + respiratory syncytial virus (RSV) was found, and 18/71 had RV/EV + multiple viral detections. Patients with single RV/EV detection required invasive mechanical ventilation (IMV) as frequently as those with RSV codetection, whereas none of those with multiple viral codetections required IMV. Species were determined in 60 samples, 58 being RV. No EV-A, EV-C, or EV-D68 were detected. RV-B and EV-B were only found in patients with other respiratory virus codetections. There were not any associations between RV/EV species and severity outcomes. To conclude, RV/EV detection alone was observed in young children with severe disease, while multiple viral codetections may result in reduced clinical severity. Differences in pathogenicity between RV and EV species could not be drawn.

## 1. Introduction

Rhinoviruses (RV) and enteroviruses (EV) are among the main causative aetiologies of lower respiratory tract infection (LRTI) in children [1]. They are RNA viruses belonging to the genus *Enterovirus* of the family *Picornaviridae*. More than 200 types of RV and EV that can infect humans are currently classified in three species of RV (RV-A, B, and C) and four of EV (EV-A to D). The clinical spectrum of RV/EV infection ranges from asymptomatic or mild symptomatic presentation to severe disease requiring respiratory support in intensive care units [1].

On the other hand, viral multiple detections are commonly observed in paediatric patients with LRTI, but there is no consensus on the relationship between viral coinfections and disease severity [2]. Most of the published literature does not focus on specific viral infections, and the role of coinfections in disease severity may be different depending on the viral aetiology [3].

The aim of this study was to evaluate the clinical impact of viral coinfection in children admitted to a paediatric intensive care unit (PICU) with LRTI and RV or EV (RV/EV) detection. Enterovirus species were also reported and analysed.

## 2. Materials and Methods

### 2.1. Subjects

Data from children aged less than 5 years with LRTI and RV/EV infection, and requiring admission to a PICU of a reference tertiary paediatric hospital (University Hospital Sant Joan de Déu, Barcelona, Spain) were prospectively collected. This medical centre provides healthcare services to a paediatric population of ≈300,000 and has a 24-bed PICU. The study period spanned from 1/2018 to 3/2020.

RV/EV disease was defined as the presence of LRTI (bronchiolitis, bronchospasm/viral wheezing, and/or pneumonia) [4] concurrent with RV/EV detection in nasopharyngeal aspirate (NPA). Suspected bacterial pneumonia was defined as the presence of fever, chest X-ray opacities, need for antibiotics, and analytical criteria (C-RP > 70 mg/L or PCT > 1 ng/mL). In our hospital, patients with respiratory failure are transferred to the PICU if they require any of the following treatments: invasive (IMV) or noninvasive (NIV) mechanical ventilation; high-flow oxygen therapy with FiO2 greater than or equal to 0.6; or if they show haemodynamic instability.

Patients with comorbidities (prematurity, immunodeficiency, congenital heart disease, and chronic lung disease—except for recurrent wheezing) were excluded. Patients were divided into three groups: RV/EV detection, RV/EV + respiratory syncytial virus (RSV) codetection with no other viral detection, and RV/EV + multiple viral codetection. Outcome variables were the need for invasive mechanical ventilation (IMV) and PICU length-of-stay (P-LOS) above the median of the overall sample (5 days).

### 2.2. Specimen Collection and Microbiological Diagnosis

NPA samples were collected from all patients with LRTI according to the normalised protocol established at the study site within the first 24 h of PICU admission. For the nasopharyngeal aspirate, a disposable catheter connected to a vacuum source was inserted into one nostril until reaching the nasopharynx. The distance from the earlobe to the tip of the patient’s nose was the length at which the catheter was inserted. Secretions were recovered into a sterile container applying suction while the catheter was drawn back. The procedure was repeated with the same catheter and container in the other nostril. Finally, three millilitres of physiological serum were suctioned and 200 μL were processed using a PCR for multiple respiratory pathogens (FilmArray-RP, BioFire Diagnostics, Salt Lake City, UT, USA), and the remaining volume was stored at −80 °C. The FilmArray-RP is a fully automated commercial test with included nucleic acid extraction and amplification for qualitative detection of the following viruses: rhinovirus/enterovirus (RV/EV), respiratory syncytial virus (RSV), parainfluenza virus, influenza virus, metapneumovirus, coronavirus, and adenovirus [5]. Since the FilmArray detection assay does not distinguish between RV and EV, an RT-nested PCR in the 5′-NC region, followed by sequencing and BLAST analysis, were subsequently performed, which allowed differentiation of RV/EV species, according to a previously published protocol [6].

### 2.3. Statistical Analysis

Data comparisons of categorical variables were performed using Pearson chi-square test or Fisher exact test. Continuous non-normal distributed variables were compared using Mann–Whitney U-test and Kruskal–Wallis analysis. A *p*-value < 0.05 was considered statistically significant. Two multivariable analyses were performed using logistic regression models and including all the variables with a cut-off point of *p* < 0.1 and “PICU stay > p50” or the “need of IMV” as the output variables. Statistical analysis was performed with SPSS v22.0 software (IBM Corp, Armonk, NY, USA).

### 2.4. Ethical Considerations

The institutional ethics board approved the study and informed consent was obtained from parents and/or legal guardians (PIC 146-17).

## 3. Results

A total of 143 patients younger than 5 years were admitted to the PICU due to RV/EV LRTI. A proportion of 68/143 had pre-existing conditions, mainly neurologic disabilities (20%) and prematurity (7%). The informed consent could not be obtained from four.

Therefore, 71 patients were included. Median age was 2.1 months (IQR: 1.2–9.3) and 40/71 were males. A total of 129 viral detections were made in these 71 children. Monthly rates of specific viral detections over the total number of detections are shown in Figure 1. In total, 57.3% of incidence of RV/EV detections occurred during the period from November to February.

RV/EV was the only virus detected in 31/71 (44%) patients. The most frequent virus codetected with RV/EV was RSV (30/71, 42%), followed by adenovirus (10/71, 14%), parainfluenza virus (9/71, 13%) and metapneumovirus (7/71, 10%). The three patient groups showed the following distribution of viruses: RV/EV alone (31/71, 44%), RV/EV + RSV detection with no other virus codetection (22/71, 31%), and RV/EV + multiple viral codetection (18/71, 25%). RSV was also detected in 8/18 (44%) patients with multiple viral codetections. No differences were found between these three cohorts in sex, race, breastfeeding status, or number of household contacts, but those patients with RV/EV detection and those with RV/EV + RSV codetection were younger than children with multiple viral codetection (*p* = 0.018). Patients with multiple viral codetection reported episodes of wheezing more often than the rest of children. Regarding other clinical variables, 30/71 (42%) patients presented with fever at hospital admission. However, children with multiple viral codetection had fever more frequently (12/18, 67%) than the other groups (*p* = 0.036) and presented symptoms for a longer time before requiring PICU admission (*p* = 0.003) (see Table 1).

RV/EV species could be determined in 60/71 positive samples. Most of the viruses detected were RV (58/60, 96.6%), these being 28/58 (48%) RV-A, 7/58 (12%) RV-B, and 23/58 (40%) RV-C. RV-B species were only detected in the RV + RSV group (*p* = 0.012) (see Table 1). Only two EV-B were detected (one in RV/EV + RSV group and one in the multiple viral codetection group). No EV-A, EV-C, or EV-D were found. Figure 2 shows the different viruses identified in each patient and Figure 3 the incidence of each RV/EV species per month.

A total of 17 out of 71 (24%) children needed IMV, but none of them had RV/EV + multiple viral codetections (*p* = 0.019). Children with RV/EV + RSV codetection underwent a longer duration of mechanical ventilation and had longer PICU and hospital stays than the other groups (see Table 1). A specific analysis based on whether RSV was detected or not is shown in Table 2. The eight patients with RSV and multiple viral codetections had a significantly shorter PICU stay (median 8 days (IQR: 5–12) vs. 3 (2–4), *p* < 0.001) and a significantly lower rate of them underwent IMV (0/8 vs. 8/22, *p* = 0.046) in comparison to those with RV/EV + RSV codetection. There were no deceased patients.

Regarding clinical severity, no differences were found in RV/EV species or other clinical/epidemiological variables, including age. A higher proportion of children with RV/EV detection + multiple viral codetections had a shorter P-LOS (15/18, *p* = 0.003) compared to the other groups (see Table 3). In the multivariable analysis, this specific group was more likely to have a shorter P-LOS (adjusted odds ratio (aOR) = 0.19 (95% CI: 0.05–0.78), *p* = 0.021). No patient with RV/EV + multiple viral codetections required IMV during the study period (0/18, *p* = 0.004), and further multivariable analysis was not considered as this was the only variable found to be associated with the outcome, with a *p*-value < 0.1.

## 4. Discussion

The role of viral coinfection in LRTI severity is controversial, with some studies reporting that viral coinfection is associated with an increased risk of PICU stay and a longer need for IMV [7], and others showing no differences in clinical outcomes on co-infected patients [8,9]. Most of the literature about viral respiratory coinfection is focused on the role of RSV infection. This report describes different clinical patterns depending on the viral codetections in the specific context of RV/EV detection in a PICU setting.

Most of the patients in whom multiple viruses were detected have had previous episodes of LRTI. Children with multiple LRTI have higher rates of codetection of pathogenic respiratory viruses with higher viral diversity and richness than children with single LRTI [10]. In this study, children with RV/EV + multiple viral codetections were reported to have a shorter PICU stay and lower need for IMV. Interestingly, patients with RV/EV + multiple viral codetections in whom RSV was also detected showed a different clinical pattern in comparison to those with the specific RV/EV + RSV codetection, having a significantly shorter PICU stay and lower need for IMV too. The immunological, metabolic, and epigenetic processes that mediate trained immunity are still widely unknown [11], but this decreased severity could be reflective of a better trained immune system in older children with previous episodes of LRTI. Despite older age and having previous episodes of wheezing not protecting children from severe outcomes in this study, detection of these multiple other viruses could correspond to traces of degrading genomes from previous events. Yet the evidence that successive viral infections are milder in children with repeated episodes of LRTI is unclear [12]. Some authors reported that RSV could cause pneumonia more often when it was associated with other viruses [13,14]. In our review, the results of this study are not in contrast with those of previous studies observing a predominance of RSV and RV/EV coinfections, since we found that patients with RV/EV + RSV underwent a longer duration of mechanical ventilation and had longer PICU and hospital stays. There are some other theories to explain these differences in clinical severity. On one hand, they could be related to the fact that different patterns of viral infection (single infection vs. coinfection with a highly pathogenic virus or coinfection with multiple viruses) may lead to different changes in respiratory microbial communities and different grades of airway inflammation induced by bacterial pathogens or viral interactions [10]. Regarding RV, RV has been positively associated with *S. pneumoniae* and *H. influenzae* colonisation in healthy infants, and RSV with *H. influenzae* [15,16]. In a prospective longitudinal cohort study of the first year of life, RV was associated with higher bacterial density and lower diversity, but these changes occurred only during symptomatic RV infections [17]. Specific associations of RV with nasal microbiotas dominated by *Moraxella* species had been associated with greater epithelial damage and inflammatory cytokine expression in asthmatic children [18]. With regard to RSV, some studies suggest that changes induced by RSV in nasopharyngeal bacterial microbiome could cause a more severe LRTI, modulating the host immune response [19]. Most of these studies have been made using qualitative PCR-based detections and analysing upper respiratory tract microbiota. Nonetheless, the effect of the respiratory tract virome and its potential effect on microbial niches in health or disease is not well known, and further research is needed to determine associations with distinct microbiome communities and infections by these pathogens. On the other hand, synergic pathways in the innate immunity produced by RSV and RV infection, and an enhanced production of RV receptor (ICAM-1) have also been proposed [20,21].

Since RV/EV can be detected in asymptomatic children [22], the clinical significance of this detection may sometimes be unclear. Pathogen-positive children are increasingly symptomatic with decreasing age [23], and RV/EV was the unique detection in very young infants with severe LRTI in our series. A similar rate of patients who underwent IMV was observed among children with RV/EV without coinfections and those with RSV coinfection, the virulence of which is beyond doubt [24]. Moreover, some authors have described that infants with RSV and RV/EV coinfection had longer hospital stays in comparison to those with RSV as the unique viral infection [25]. In addition, recent data clearly show that viral aetiology of LRTI could be a factor that determines short- and long-term outcomes [26]. In this way, RV infection in young children is associated with atopic predisposition and high risk of asthma, while RSV infection seems not to be related with asthma or atopy development [27]. All these results reinforce the role of RV/EV in LRTI in a clinical scenario of a very young infant in whom this virus is found. RV viral load in the upper respiratory tract has been analysed as a potential biomarker to distinguish between symptomatic patients with respiratory viral infection and asymptomatic carriers of the virus, with heterogenous results [28].

Regarding the RV and EV species that were identified in our study, almost all were RV. EV-D68 was not found in any patient, despite being detected in a previous series published by our group [29]. Proportions of RV species in children with LRTI were like those previously described elsewhere [30], with an RV-A and C predominance. RV species have different patterns of circulation. Infections occur all year round, but two peaks of infection are classically reported in northern hemisphere countries, the first between April and May, and the second between September and October [28,31]. RV-C could demonstrate a different trend, with a peak of infection during winter months [32]. In our study, RV-C was detected almost the whole year, similarly to others’ results [33]. On the other hand, EV-D68, an EV type most frequently associated to respiratory diseases, has been described to be more pathogenic than most RV species [34], in addition to causing acute flaccid paralysis. The fact that no EV-D68 species was detected could be due to the epidemiologic circulation pattern of this EV type, which alternates years of high incidence with years of low circulation [35]. EV-B was only found in two patients, but this species had been rarely observed in LRTI [29,30,36]. Furthermore, EV-B was detected in patients with other viral multiple coinfections, so its pathogenic implication is not clear, and may result in asymptomatic infection. This emphasises the need to improve diagnostic testing strategies for respiratory viruses to successfully distinguish between RV and EV, viruses that belong to the same genus but with different biological and clinical characteristics. Finally, evidence that members of RV-C species might be more virulent [36,37] continues to be controversial. No differences in outcome variables between RV species were found in this study, similarly to others’ results showing similar rates of ICU admission, need for IMV, and length of hospital stay between RV species [38], and no associations between a given RV species and the type of infection (upper respiratory tract, LRTI, or protracted infection) [39].

The main limitations of this study are its observational design and the conduct of the study in a single setting, so its results and conclusions could have limited external generalisability. Second, the small sample size of the groups does not allow the establishment of firm conclusions about the absence of differences between them. Finally, although upper respiratory microbiological screening with qualitative PCR detection may not be conclusive, these specimens (NPA or nasal/throat swabs) are the most widely used in paediatric clinical settings [14]. Obtaining sputum or bronchoalveolar lavage is hardly applicable. Nonetheless, literature comparing viral detections in NPA and bronchoalveolar lavage shows a high correlation between them in immunocompetent children [40]. Quantitative microbiological detection is uncommonly reported in clinical daily practice due to severe limitations in interpreting the results [41].

To conclude, the presence of RV/EV with other respiratory virus codetection appears not to be decisive to cause severe LRTI in young children. RV/EV monoinfection and RSV coinfection had similar rates of infants undergoing IMV, whereas multiple viral coinfection was associated with less clinical severity. Differences in pathogenicity between RV and EV species could not be drawn from this study, despite the fact that RV-B and EV-B were only found to be associated with other respiratory viral infections.

## Figures and Tables

**Figure 1 viruses-13-02059-f001:**
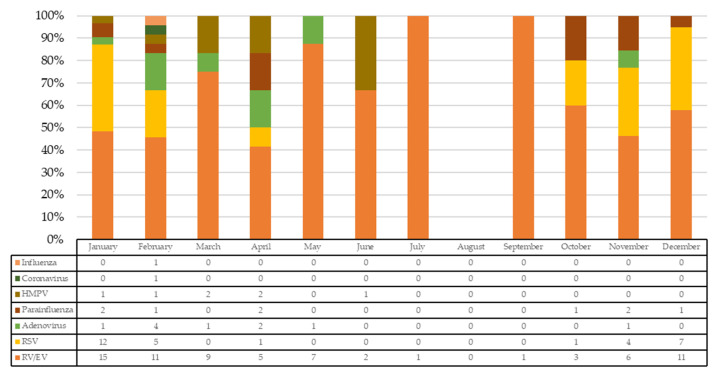
Number and rate of specific viral detections over the total number of detections. HMPV: human metapneumovirus; RSV: respiratory syncytial virus; RV/EV: rhinovirus/enterovirus.

**Figure 2 viruses-13-02059-f002:**
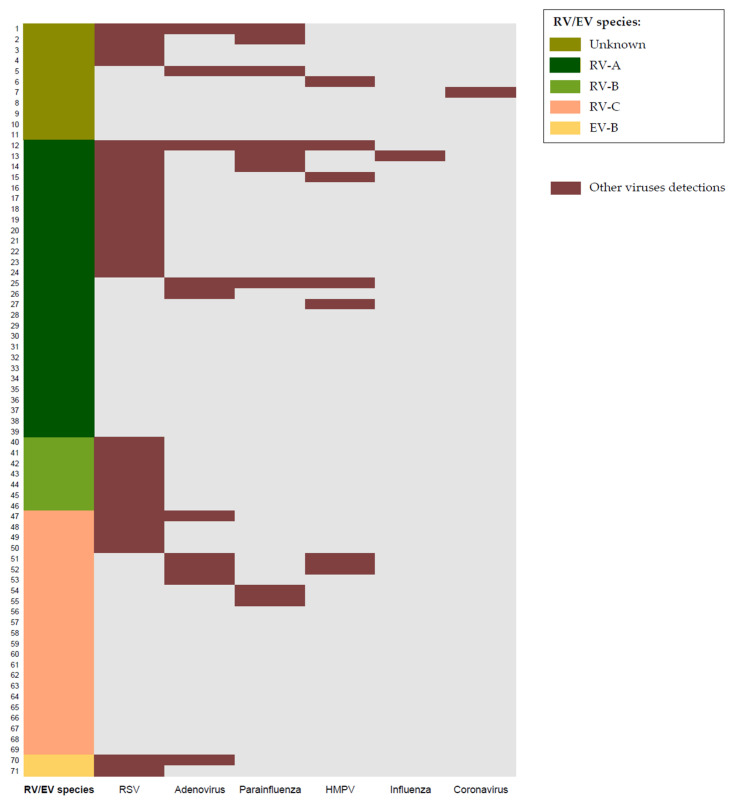
Viral detections and EV/RV species in each patient. MPV: human metapneumovirus; RSV: respiratory syncytial virus; RV/EV: rhinovirus/enterovirus.

**Figure 3 viruses-13-02059-f003:**
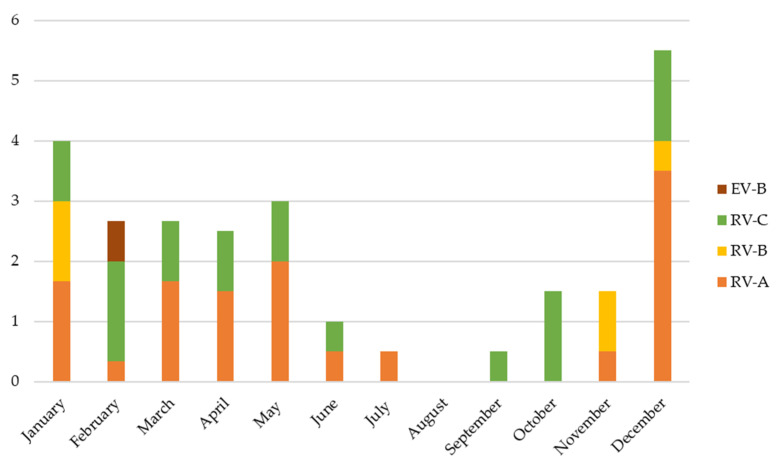
Number of detections of each EV/RV species per month.

**Table 1 viruses-13-02059-t001:** Demographic, clinical, and microbiological characteristics of children with RV/EV detection, RV/EV + RSV codetection, and RV/EV + multiple viral codetection.

Variables	Total *n* = 71	RV/EV *n* = 31	RV/EV + RSV Codetection *n* = 22	RV/EV + Multiple Viral Codetection *n* = 18	*p*-Value *
Sex (male), *n* (%)	40 (56%)	17 (55%)	16 (73%)	7 (39%)	0.097
Age (months), median (IQR)	2.1 (1.2–9.3)	1.7 (0.7–8.1)	1.8 (1.0–2.7)	7.9 (2.2–22.7)	0.018
Race (Caucasian), *n* (%)	51 (78%)	20 (64%)	19 (86%)	12 (67%)	0.187
Breastfeeding, *n* (%)	57 (80%)	27 (87%)	17 (77%)	13 (72%)	0.412
Parental smoking, *n* (%)	23 (32%)	12 (39%)	7 (33%)	4 (21%)	0.492
Household contacts, *n* (%)	4 (4–4)	4 (4–4)	4 (3–5)	4 (4–4)	0.926
Recurrent wheezing, *n* (%)	28 (40%)	11 (37%)	4 (18%)	13 (72%)	0.002
Fever, *n* (%)	30 (42%)	9 (29%)	9 (41%)	12 (67%)	0.036
Days with symptoms before PICU admission, median (IQR)	3 (1–5)	2 (1–4)	3 (2–5)	5 (3–6)	0.003
Length of PICU stay (days), median (IQR)	5 (3–9)	6 (3–9)	8 (4–12)	3 (2–5)	0.004
Hospital stay (days), median (IQR)	11 (9–18)	12 (9–20)	13 (10–17)	9 (6–11)	0.028
Chest X-ray (total *n* = 60)					0.572
Normal, *n* (%)	17 (28%)	8 (32%)	5 (29%)	4 (22%)	
Chest X-ray opacities 1 quadrant, *n* (%)	18 (30%)	10 (40%)	4 (23%)	5 (28%)	
Chest X-ray opacities > 1 quadrant, *n* (%)	24 (40%)	7 (28%)	8 (47%)	9 (50%)	
NIVM, *n* (%)	67 (94%)	29 (93%)	21 (95%)	17 (94%)	0.957
IMV, *n* (%)	17 (24%)	9 (29%)	8 (36%)	0 (0%)	0.019
Days of MV, median (IQR)	4 (3–8)	4 (2–7)	8 (3–11)	3 (1–4)	0.002
Total white blood cell count (cells × 10^9^/L), median (IQR)	11.6 (8.2–16.4)	11.8 (8.2–17.1)	9.4 (6.8–13.4)	12.9 (9.0–19.3)	0.265
Neutrophils (cells × 10^9^/L), median (IQR)	5.2 (2.7–8.3)	5.2 (2.5–8.1)	4.4 (1.9–8.5)	5.4 (3.8–8.0)	0.516
C-RP (mg/L), median (IQR)	33.9 (13.2–66.9)	37 (6.9–73)	30 (15–52)	34 (18–71)	0.962
PCT (ng/mL), median (IQR)	0.33 (0.14–1.67)	0.40 (0.12–1.08)	0.22 (0.19–2.78)	0.54 (0.13–2.13)	0.770
Suspected bacterial pneumonia criteria, *n* (%)	14 (23%)	5(20%)	3 (18%)	6 (33%)	0.480
RV/EV species:					
-RV-A	28	12 (39%)	9 (41%)	7 (39%)	0.986
-RV-B	7	0 (0%)	7 (32%)	0 (0%)	<0.001
-RV-C	23	14 (45%)	3 (14%)	6 (33%)	0.054
-EV-B	2	0 (0%)	1 (4%)	1 (6%)	0.442
-Unknown	11	5 (16%)	2 (9%)	4 (22%)	0.517

C-RP: C-reactive protein; IMV: invasive mechanical ventilation; IQR: interquartile range; MV: mechanical ventilation; NIVM: non-invasive mechanical ventilation; NPA: nasopharyngeal aspirate; PCT: procalcitonin; PICU: Paediatric Intensive Care Unit; RSV: respiratory syncytial virus; RV/EV: rhinovirus/enterovirus. * Comparisons between categorical variables were performed using Pearson chi-square test. Fisher exact test was used if any expected count was < 5. Continuous variables were compared using Kruskal–Wallis analysis.

**Table 2 viruses-13-02059-t002:** Main demographic and clinical characteristics of children with RV/EV infection, according to RSV detection.

Variables	RV+ RSV– (*n* = 41)	RV+ RSV+ (*n* = 30)	*p*-Value *	RV + RSV as the Sole Codetection (*n* = 22)	RV + RSV in Codetection with Other Viruses (*n* = 8)	*p*-Value *
Age (months), median (IQR)	2.9 (1.2–13.1)	1.6 (1.1–5-5)	0.625	1.9 (1.0–2.7)	8.7 (1.7–18.6)	0.090
Recurrent wheezing, *n* (%)	9 (22.5%)	6 (20%)	0.801	2 (9%)	4 (26%)	0.029
Fever, *n* (%)	16 (39%)	14 (47%)	0.520	9 (41%)	5 (62%)	0.417
Days with symptoms before PICU admission, median (IQR)	2 (1–4)	3 (2–5)	0.006	3 (2–5)	6 (4–11)	0.008
Length of PICU stay (days), median (IQR)	5 (2–8)	5 (3–9)	0.512	7.5 (4.7–12.0)	3 (2.2–3.7)	<0.001
Hospital stay (days), median (IQR)	11.5 (8–18)	11 (9–15)	0.863	13 (10.0–17.2)	9.5 (6.7–10.7)	0.060
IMV, *n* (%)	9 (22%)	8 (27%)	0.646	8 (36%)	0 (0%)	0.046
Days of MV, median (IQR)	4 (2–6)	5 (3–9)	0.267	8 (3–11)	3 (1–3)	0.002

* Comparisons between categorical variables were performed using Pearson chi-square test and Fisher exact test was used if any expected count was < 5. Continuous variables were compared using the Mann–Whitney U-test.

**Table 3 viruses-13-02059-t003:** Outcome variables: PICU length of stay and need for invasive mechanical ventilation.

	Univariate Analysis			Multivariable Analysis *		Univariate Analysis		
Variables	PICU Stay > 5 d (*n* = 33)	PICU Stay ≤ 5 d (*n* = 38)	*p*-Value **	Adjusted Odds-Ratio	*p*-Value	IMV (*n* = 17)	NIMV (*n* = 54)	*p*-Value **
Sex (male), *n* (%)	20 (61%)	20 (53%)	0.499			10 (56%)	30 (56%)	0.813
Age (months), median (IQR)	2.2 (1.2–5.6)	2.1 (0.9–14.8)	0.836			1.8 (1.0–9.9)	2.1 (1.2–10.3)	0.652
Recurrent wheezing, *n* (%)	10 (30%)	18 (47%)	0.170			6 (35%)	22 (41%)	0.649
Race:								
Caucasian, *n* (%)	25 (76%)	26 (68%)	0.493			13 (76%)	38 (70%)	0.625
Others, *n* (%)	8 (24%)	8 (32%)				4 (24%)	16 (30%)	
Fever, *n* (%)	15 (45%)	15 (39%)	0.611			8 (47%)	22 (41%)	0.646
Chest X-ray at hospital admission (total *n* = 60)								
Normal, *n* (%)	4 (15%)	13 (38%)	0.110			1 (8%)	16 (33%)	0.183
Chest X-ray opacities 1 quadrant, *n* (%)	11 (42%)	8 (23%)				4 (33%)	15 (31%)	
Chest X-ray opacities > 1 quadrant, *n* (%)	11 (42%)	13 (38%)				7 (58%)	17 (35%)	
Viral detections:								
RV/EV, *n* (%)	16 (48%)	15 (39%)	0.445	-	-	9 (53%)	22 (41%)	0.376
RV/EV + RSV, *n* (%)	14 (42%)	8 (21%)	0.052	1.64 (0.54–5.02)	0.386	8 (47%)	14 (26%)	0.101
RV/EV + Multiple viral codetection, *n* (%)	3 (9%)	15 (40%)	0.003	0.19 (0.05–0.78)	0.021	0 (0%)	18 (33%)	0.004
RV/EV species								
RV A, *n* (%)	14 (42%)	14 (37%)	0.631			6 (35%)	22 (41%)	0.689
RV B, *n* (%)	4 (12%)	3 (8%)	0.697			1 (6%)	6 (11%)	0
RV C, *n* (%)	8 (24%)	15 (39%)	0.171			5 (29%)	18 (33%)	0.763
EV B, *n* (%)	1 (3%)	1 (3%)	1			1 (6%)	1 (2%)	0.424
Unknown, *n* (%)	6 (18%)	5 (13%)	0.560			4 (24%)	7 (13%)	0.441
Total white blood cell count (cells × 10^9^/L), median (IQR)	10,500 (7425–15,825)	12,000 (8600–17,350)	0.296			9300 (6600–15,400)	11,750 (8925–17,000)	0.316
Neutrophils (cells × 10^9^ /L), median (IQR)	4490 (2675–8375)	5450 (3850–7850)	0.494			4500 (1800–9000)	5200 (3800–7700)	0.794
C-RP (mg/L), median (IQR)	38 (21–64)	30 (11–72)	0.302			32 (12–63)	35 (13–66)	0.946
PCT (ng/mL), median (IQR)	0.35 (0.19–1.49)	0.20 (0.08–2.08)	0.328			0.22 (0.18–2.78)	0.35 (0.12–1.34)	0.872

C-RP: C-reactive protein; IQR: interquartile range; NPA: nasopharyngeal aspirate; PCT: procalcitonin; PICU: Paediatric Intensive Care Unit. * Logistic regression model using the “enter” method. All the variables with a univariate *p*-value < 0.1 were introduced in the model. Hosmer–Lemeshow *p*-value = 1. ** Comparisons between categorical variables were performed using Pearson chi-square test and Fisher exact test was used if any expected count was < 5. Continuous variables were compared using the Mann–Whitney U-test.

## Data Availability

The datasets generated and/or analysed during the current study are available from the corresponding author on reasonable request.

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
