# Peer review of "Lower Respiratory Tract Infection and Genus Enterovirus in Children Requiring Intensive Care: Clinical Manifestations and Impact of Viral Co-Infections"

_viruses, 2021, doi:10.3390/v13102059_

Round 1
Reviewer 1 Report
In this study the authors investigate the presence and clinical impact of respiratory viruses including rhinovirus/enterovirus species in children with lower respiratory tract infections. The authors observed that children with rhinovirus and RSV monoinfectioins required invasive mechanical ventilation whereas children with multiple viruses did not. Species of the rhinovirus B and enterovirus B genus were only detected in co-occurrence with other viruses. Overall, no associations were observed between rhinovirus or enterovirus and severity outcomes.
General comments: The study could be substantially enhanced by the addition of data that I have specifically addressed. Methodology is poorly described when it comes to the NPA specimen processing and the virus detection assay. The authors state that they wish to focus on RV/EV presence. However, the possibility of RSV-driven infections is high and the authors should take this into consideration and re-evaluate the statistical analysis and relevant discussion. The authors describe LRTIs, however, they have sampled the upper respiratory tract. How is this justified? To what extent their data could be biased towards virus presence in the upper system and with no relation whatsoever to the disease outcomes? A more extensive review of the literature is advised and inclusion of a higher number of references.
Specific comments:
Line 64: Provide a copy of the nasopharyngeal specimen collection protocol or a reference.
Line 66: Provide a reference for the virus detection assay used in the study.
Line 66: Provide details of the nasopharyngeal specimen processing protocol in a separate subsection. This should include specimen processing, nucleic acid isolation and description of the virus detection assay.
For example, is this a quantitative method that provides output as occurrence data? Are there raw Ct and melting curve analysis data available (provide figures)? How are these combined? Regarding the virus detection assay, it is stated that it cannot discriminate amongst different virus species (or strains). However, different species due to their genetic heterogeneity in the PCR amplified regions are expected to produce different melting profiles. How are these evaluated as a species-specific positive (or negative) result? How is virus occurrence determined? Are there positive and negative controls? Did the authors include replicate reactions in the analysis? A fairly good example of a detailed description of virus assay quantitative data analysis can be found in The Journal of Infectious Diseases, Volume 213, Issue 6, 15 March 2016, Pages 915–921, https://doi.org/10.1093/infdis/jiv513.
Line 71: Re-sequencing of the rhinovirus/enterovirus positive samples was performed. The authors should include a list of the identified virus strains followed by alignment and sequence similarity cladograms. The authors should present the frequency of detection for each virus strain and analyse these data in relation to the LRTI severity outcomes.
Line 72: The variables used to define LRTI should be used in the analysis, i.e. bronchiolitis, bronchospasm, and wheezing.
Line 101 and onwards: Please provide %incidence of virus detection along with the absolute counts. Include detection data (incidence and absolute number) of RSV positive samples. To this end, given the median age of the children, the known ability of RSV to infect the lower respiratory tract, and the seasonal circulation of RSV which coincides with patient inclusion, I would expect most of the observed LRTIs to be caused by RSV. The authors should comment on the above. If the RSV incidence is high, then the study should be re-evaluated as it is likely that it describes a potential effect of non-RSV virus presence in RSV induced LRTIs.
Provide a figure with stacked bar plots presenting the different viruses identified in each sample. A second similar plot should focus on RV/EV strains identified through re-sequencing.
Statistical analysis should evaluate RSV presence and disease severity. Without this it is hard to evaluate any of the statistical data presented in this study as the cofounding effect of RSV is neglected.
Line 124. Was RV-C presence related to more severe LRTIs? The authors should discuss about RV species A, B, and C circulation and presence in LRTIs.
Line 169-173: The statement is too vague. Moreover, I do not think that we know so much about the effect of the eukaryotic virome that is present in the respiratory tract and its potential effect on microbial niches in health or disease. Causative effects cannot be drawn by PCR-based virus detection. Especially, in the context of this study where the virus data are only qualitative. Moreover, the authors do not provide evidence of virion presence in their samples, or assembled virus genomes, so any discussion about functionality is vague and with poor scientific impact.
Line 176: Is this true?
Line 190-191: I do not think this is controversial. Please expand.
Line 192: Add the absence of virus quantitative data.
Author Response
General comments
- The possibility of RSV-driven infections is high and the authors should take this into consideration and re-evaluate the statistical analysis and relevant discussion.
A new Table (new Table 2) has been added to the manuscript comparing RSV- and RSV+ infections. With regard RSV, specific comments made by the reviewer are answered below.
- The authors describe LRTIs, however, they have sampled the upper respiratory tract. How is this justified? To what extent their data could be biased towards virus presence in the upper system and with no relation whatsoever to the disease outcomes?
We agree with the reviewer, and we have used the word “detection” rather than “infection” along all the manuscript. On the other hand, sputum or bronchoalveolar lavage are very difficult to obtain in infants, so providing these results may have a low clinical impact. A new reference has been added (ref 12).
We agree that upper respiratory microbiological screening may not be conclusive when PCR-based results are reported in an individual. Despite it remains possible that some of the children with RV in our series were asymptomatic or in recovery; overall, our data suggest that in children with severe bronchiolitis, RV plays a central role in the clinical course and is not asymptomatic. The discussion has been expanded:
Page 10, line 298: “Finally, although upper respiratory microbiological screening with qualitative PCR detection may not be conclusive, these specimens (NPA or nasal/throat swabs) are the most widely used in paediatric clinical settings [12]. Obtaining sputum or bronchoalveolar lavage is hardly applicable.”
Finally, there are studies evaluating the correlation between upper and lower respiratory tract diagnosis using PCR approaches, but the heterogeneity in the included subjects’ characteristics is very high. We have added a new reference with evidence to ensure that nasopharyngeal samples could be sufficient to the diagnosis of lower respiratory tract viral infection in immunocompetent children (ref 37, Rev Fr Allergol 2013;53:59-64):
Page 10, line 302: “Nonetheless, literature comparing viral detections in NPA and bronchoalveolar lavage shows a high correlation between them in immunocompetent children [38].”
Specific comments:
- Provide a copy of the nasopharyngeal specimen collection protocol or a reference. Provide a reference for the virus detection assay used in the study. Provide details of the nasopharyngeal specimen processing protocol in a separate subsection. This should include specimen processing, nucleic acid isolation and description of the virus detection assay.
We have modified the methods section adding more details about the specimen processing and description of virus detection assay. We updated the references and we have also added the suggested subsections:
Page 2, line 80: “Nasopharyngeal aspirate (NPA) samples were collected from all patients with LRTI according to the normalized protocol established at the study site within the first 24 hours of PICU admission. For the nasopharyngeal aspirate, a disposable catheter connected to a vacuum source was inserted into one nostril until reaching the nasopharynx. The distance from the earlobe to the tip of the patient’s nose was the length at which the catheter was inserted. Secretions were recovered into a sterile container applying suction while the catheter was drawn back. The procedure was repeated with the same catheter and container in the other nostril. Finally, three milliliters of physio-logical serum were suctioned and 200μL were processed using a PCR for multiple respiratory pathogens (FilmArray-RP, BioFire Diagnostics, US) and the remaining volume was stored at −80°C. The FilmArray-RP is a fully automated commercial test with included nucleic acid extraction and amplification for qualitative detection of the fol-lowing viruses: rhinovirus/enterovirus (RV/EV), Respiratory Syncytial Virus (RSV), parainfluenza virus, influenza virus, metapneumovirus, coronavirus, adenovirus [5]. Since the FilmArray detection assay does not distinguish between RV and EV, a RT-nested PCR in the 5'-NC region followed by sequencing and BLAST analysis, were subsequently performed, which allowed differentiation of RV/EV species, according to a previously published protocol [6].”
For example, is this a quantitative method that provides output as occurrence data?
We have clarified that FA-RP is a fully automated qualitative method. A reference has also been added to clarify it (ref 5, Expert Rev Mol Diagn 2013; 13: 779 - 288).
Are there raw Ct and melting curve analysis data available (provide figures)? How are these combined? Regarding the virus detection assay, it is stated that it cannot discriminate amongst different virus species (or strains). However, different species due to their genetic heterogeneity in the PCR amplified regions are expected to produce different melting profiles. How are these evaluated as a species-specific positive (or negative) result? How is virus occurrence determined?
FilmArray is a commercial method with IVD-CE certification for simultaneous qualitative detection and identification of multiple respiratory virus. Using endpoint melting curve data, the BioFire System software automatically analyzes the results for each target on the panel. The system has their own algorithm and raw CT and melting curves are not available for the user.
Are there positive and negative controls?
The system includes an internal control as quality control in each determination. In addition, as accredited laboratory we participate in different external quality programs which included negative and positive controls of all tests used in our lab for microbial diagnosis. The FilmArray system is also included in this external controls with excellent results.
Did the authors include replicate reactions in the analysis?
No. As an IVD-CE test used for routine diagnosis no replication reactions are needed.
A fairly good example of a detailed description of virus assay quantitative data analysis can be found in The Journal of Infectious Diseases, Volume 213, Issue 6, 15 March 2016, Pages 915–921, https://doi.org/10.1093/infdis/jiv513.
Thanks for the suggestion. We have followed the example.
- Re-sequencing of the rhinovirus/enterovirus positive samples was performed. The authors should include a list of the identified virus strains followed by alignment and sequence similarity cladograms. The authors should present the frequency of detection for each virus strain and analyze these data in relation to the LRTI severity outcomes.
EV/RV were detected using a commercial real-time PCR assay which does not distinguish between EV and RV (both viruses belong to the same genus and the PCR was designed in a high conserved region of the genome). In positive samples, a fragment of the 5`-non-coding region was amplified used a conventional RT-PCR. Sequencing of this region only allows to determine the species of EV or RV, but not the specific serotype. This has been clarified in the revised manuscript:
Page 3, line 90: “Since the FilmArray detection assay does not distinguish between RV and EV, a RT-nested PCR in the 5'-NC region followed by sequencing and BLAST analysis, were subsequently performed, which allowed differentiation of RV/EV species, according to a previously published protocol [6].”
Unfortunately, the specific genotyping of the EV and RV could not be achieved. Then, a list of the identified virus strains followed by alignment and sequence similarity cladograms cannot be provided, neither the frequency of detection for each virus strain in relation to the LRTI severity outcomes.
We want to remark that RV and EV species had been considered as potential variables associated with severity (old Table 2, new Table 3).
- The variables used to define LRTI should be used in the analysis, i.e. bronchiolitis, bronchospasm, and wheezing.
Bronchiolitis, bronchospasm/viral-wheezing and/or pneumonia were the clinical entities used to define the inclusion criteria. Distinguishing these entities from each other is very difficult, due to the heterogeneity in the definitions. Moreover, they are not mutually exclusive and can overlap between them (Douros K and Everard ML (2020) Time to Say Goodbye to Bronchiolitis, Viral Wheeze, Reactive Airways Disease, Wheeze Bronchitis and All That. Front. Pediatr. 8:218). Thus, we decided to group them as lower respiratory tract infections (a definition that is less controversial), but at the same time we offered information on whether children had had previous episodes of bronchospasm/wheezing and their age ranges (Table 1). We believe that all this information can help to get a more accurate idea of what type of patients have been included in the study, as some scientific societies stablish the separation between bronchiolitis and bronchospasm according to this information. On the other hand, the categorization of “pneumonia” was already included in table 1 when we provide information on the number of patients with chest-X-ray opacities.
We have added a reference with LRTI definition (ref 4, Front Pediatr 2020; 8: 218):
Page 2, line 65: “RV/EV disease was defined as the presence of LRTI (bronchiolitis, bronchospasm/viral-wheezing and/or pneumonia) [4] concurrent with RV/EV detection in NPA.”
- Please provide %incidence of virus detection along with the absolute counts.
This data has been added using a Figure (new figure 1).
- Include detection data (incidence and absolute number) of RSV positive samples.To this end, given the median age of the children, the known ability of RSV to infect the lower respiratory tract, and the seasonal circulation of RSV which coincides with patient inclusion, I would expect most of the observed LRTIs to be caused by RSV. The authors should comment on the above. If the RSV incidence is high, then the study should be re-evaluated as it is likely that it describes a potential effect of non-RSV virus presence in RSV induced LRTIs.
Information regarding RSV virological detection has been added in the manuscript. We provide a new table (RSV vs no-RSV) with the main variables used in the comparisons of the three main cohorts. The included variables are those with statistically significant differences in Table 1, which include the main outcomes -P-LOS and IMV-. See new Table 2. The new table helps the reader to understand the initial approach of the analysis (the 3 previously stablished cohorts): Patients with RV+RSV as the unique detections are clearly different in clinical course and demographic characteristics in comparison to those with multiple viral co-detections, thus this last group had been combined with that of the patients with multiple viral co-infections in Table 1 (multiple viral coinfected patients, regardless of whether RSV is one of the detections or not).
RSV has a very good correlation between NPA detection and pathogenicity (Rev Fr Allergol 2013;53:59-64) and on the other hand it is very rarely found in non-symptomatic individuals (Pediatrics. 2014 Nov;134(5):e1474-502). We want to underline that the inclusion period included 2 whole years, not only RSV epidemic peaks. The new Figure 1 helps to clarify it.
- Provide a figure with stacked bar plots presenting the different viruses identified in each sample. A second similar plot should focus on RV/EV strains identified through re-sequencing.
A new Figure (Figure 2) was added according to this suggestion.
- Statistical analysis should evaluate RSV presence and disease severity. Without this it is hard to evaluate any of the statistical data presented in this study as the cofounding effect of RSV is neglected.
See point 5.
- Was RV-C presence related to more severe LRTIs? The authors should discuss about RV species A, B, and C circulation and presence in LRTIs.
RV-C was reported previously to be the most common RV species in wheezing children and the major pathogen in severe wheezing and febrile respiratory illness in young children, but this is not a constant observation in different series. This issue is commented below.
A new paragraph about RV species, circulation and differences in pathogenicity was added to the revised manuscript according to this suggestion. Moreover, we provided the incidence per month of each RV/EV species to discuss the results with existing literature (Figure 2):
Page 10, line 270: “Regarding the RV and EV species that were identified in our study, almost all were RV. EV-D68 was not found in any patient despite being detected in a previous series published by our group [27]. Proportions of RV species in children with LRTI were like those previously described in other Spanish series [28], with a RV-A and C predominance. RV species have different patterns of circulation. Infections occur all year round, but two peaks of infection are classically reported in northern hemisphere countries, the first between April and May and the second between September and October [26, 29]. RV-C could demonstrate a different trend, with a peak of infection during winter months [30]. In our study, RV-C was detected almost the whole year, similarly to others’ results [31]. On the other hand, EV-D68, an EV type most fre-quently associated to respiratory diseases has been described to be more pathogenic than most of RV species [32], in addition to causing acute flaccid paralysis. The fact that no EV-D68 species was detected could be due to the epidemiologic circulation pattern of this EV type, which alternates years of high incidence with years of low circulation [33]. EV-B were only found in two patients, but this species had been rarely observed in LRTI [27,34]. Furthermore, EV-B was detected in patients with other viral multiple co-infections, so their pathogenic implication is not clear, and may result in asymptomatic infection. This emphasizes the need to improve diagnostic testing strategies for respiratory viruses to successfully distinguish between RV and EV, viruses that belong to the same genus but with different biological and clinical characteristics.”
Seasonal circulation of each RV/EV species as also been added in the new Figure 3.
The statement is too vague. Moreover, I do not think that we know so much about the effect of the eukaryotic virome that is present in the respiratory tract and its potential effect on microbial niches in health or disease. Causative effects cannot be drawn by PCR-based virus detection. Especially, in the context of this study where the virus data are only qualitative. Moreover, the authors do not provide evidence of virion presence in their samples, or assembled virus genomes, so any discussion about functionality is vague and with poor scientific impact.
When considering the airway microbiome, it is impossible to exclude the impact of respiratory infections, especially viral infections, on the composition of the bacteria present. However, it is difficult to delineate if respiratory bacterial composition creates an environment that is more suitable for viral infection and thus increases the susceptibility to, duration of, or severity of an acute respiratory viral infection, or, if, conversely, viral infections lead to changes in microbial composition that subsequently cause an increase in pathogenic bacterial and/or fungal infections. Indeed, there is evidence that both situations may occur. Most studies that have investigated respiratory viral infections and the microbiome have focused on RSV and RV.
According to this suggestion, and to elaborate the discussion about the relationship between these viruses, microbiome and clinical severity, we have expanded that paragraph on the revised manuscript. We have also added other possible explanations to our observations. We let the reviewer and the editors to decide if it is more useful now or we should delete it:
Page 9, line 233: “There are some theories to explain these observations. On one hand, these differences in clinical severity could be related to the fact that different patterns of viral infection (single infection vs co-infection with a highly pathogenic virus or co-infection with multiple viruses) may lead to different changes in respiratory microbial communities and different grades of airway inflammation induced by bacterial pathogens or viral interactions [10]. Regarding RV, RV has been positively associated with S. pneumoniae and H. influenzae colonization in healthy infants, and RSV with H. influenzae [13,14]. In a prospective longitudinal cohort study of the first year of life, RV was associated with higher bacterial density and lower diversity, but these changes occurred only during symptomatic RV infections [15]. Specific associations of RV with nasal microbiotas dominated by Moraxella species had been associated with greater epithelial damage and inflammatory cytokine expression in asthmatic children [16]. With regard RSV, some studies suggest that changes induced by RSV in nasopharyngeal bacterial microbiome could cause a more severe LRTI, modulating the host immune response [17]. Most of these studies have been made using qualitative PCR-based detections and analyzing upper-respiratory tract microbiota. Nonetheless, the effect of the respiratory tract virome and its potential effect on microbial niches in health or disease is not known so much and further research is needed to determine associations with distinct microbiome communities and infections with these pathogens. On the other hand, synergic pathways in the innate immunity produced by RSV and RV infection, or and enhanced production of RV receptor (ICAM-1) had also been proposed [18,19].”
- I do not think this is controversial. Please expand.
As it was previously reported, EV-B species had been rarely observed causing LRTI (see References 27, 28 and 34). In our study these species were only found in two patients, and these detections were made in patients with RSV and with other viral multiple co-detections. For that reason, the pathogenic implication of EV-B species is not clear, and may result in asymptomatic infection
Regarding RV-C and its highest pathogenicity issue we have expanded the discussion with new references to clarify it:
Page 9, line 289: “Finally, evidence that members of RV-C species might be more virulent [34-35] continues being controversial. No differences in outcome variables between RV species were found in this study, similarly to others’ results showing similar rates of ICU admission, need of IMV and length of hospital stay between RV species infections [36], and no associations between a given RV species and the type of infection (upper-respiratory tract, LRTI, or protracted infection) [37].”.
- Add the absence of virus quantitative data.
A new statement has been done to remark this limitation and expanded with new references:
Page 10, line 277: “Finally, although upper respiratory microbiological screening with qualitative PCR detection may not be conclusive, these specimens (NPA or nasal/throat swabs) are the most widely used in paediatric clinical settings [12]. Obtaining sputum or bronchoalveolar lavage is hardly applicable. Nonetheless, literature comparing viral detections in NPA and bronchoalveolar lavage shows a high correlation between them in immunocompetent children [38]. Quantitative microbiological detection is uncommonly reported in clinical daily practice due to severe limitations in interpreting the results [39].”
We think that the present version of manuscript includes the answers to the main questions of the reviewers and we hope that it could be accepted for publication in Viruses.
I will be looking forward to hearing from you.
Sincerely yours,
Cristian Launes, MD, MSc, PhD
Servei de Pediatria. Grup de Recerca en Malalties Infeccioses. CIBERESP.
Professor Associat. Hospital Sant Joan de Déu (Universitat de Barcelona)
Pg Sant Joan de Déu,2
08950 Esplugues (Barcelona) Spain
Tel 93 253 21 40 / Fax 93 600 94 60
www.sjdhospitalbarcelona.org
ORCID ID: http://orcid.org/0000-0002-0913-9303
Reviewer 2 Report
This report may not completely demonstrate that the relationship between RV/EV species and other viruses co-infection in young children with genus Enterovirus severe LRTI. This study conclude that there were not associations between any RV/EV species and severity outcomes just from the proportion of different kinds of co-infection, which may be overstatement. Therefore, the results obtained so far are not sufficient to support the existing conclusions. In addition, some sentences in this article are vague and difficult to understand. And extensive editing of English language and style required.
Author Response
- This report may not completely demonstrate that the relationship between RV/EV species and other viruses co-infection in young children with genus Enterovirus severe LRTI. This study conclude that there were not associations between any RV/EV species and severity outcomes just from the proportion of different kinds of co-infection, which may be overstatement. Therefore, the results obtained so far are not sufficient to support the existing conclusions.
We do not completely understand this comment. RV and EV species had been analyzed as risk factors for a more severe disease in old Table 2 (new Table 3). Concretely:
|
Variables |
|
PICU stay > 5 d
|
PICU stay < 5 d
|
P-value |
|
IMV (n=17) |
NIMV (n=54) |
P value |
|
|
RV/EV species RV A, n (%) RV B, n (%) RV C, n (%) EV B, n (%) Unknown, n (%) |
14 (42%) 4 (12%) 8 (24%) 1 (3%) 6 (18%) |
14 (37%) 3 (8%) 15 (39%) 1 (3%) 5 (13%) |
0.631 0.697 0.171 1 0.560 |
|
6 (35%) 1 (6%) 5 (29%) 1 (6%) 4 (24%) |
22 (41%) 6 (11%) 18 (33%) 1 (2%) 7 (13%) |
|
0.689 1 0.763 0.424 0.441 |
|
As none of them were associated with the outcomes with a p<0.1 they were not considered for the multivariable model. Statistical procedures were revised by an expert in statistical methodology.
Moreover, there is very scarce literature focusing on the objectives of this study (RV/EV detection and the role of RV/EV species and other virus coinfection) in a PICU setting. We used procedures and microbiological techniques that are easily available and used widely. Thus, this proximity to the real clinical daily practice makes our results of high interest for other researchers and clinicians.
- In addition, some sentences in this article are vague and difficult to understand. And extensive editing of English language and style required.
English language and style have been revised.
We think that the present version of manuscript includes the answers to the main questions of the reviewers and we hope that it could be accepted for publication in Viruses.
I will be looking forward to hearing from you.
Sincerely yours,
Cristian Launes, MD, MSc, PhD
Reviewer 3 Report
The article describes a cohort study that evaluates the clinical impact of RV/EV species and other viruses co-infection in children with LRTI. Three groups were included in the study: RV/EV infection, RV/EV + RSV co-infection with no other viral detection, and RV/EV + multiple viral co-infection. A similar rate of children who needed IMV was observed among RV/EV mono-infection and RSV co-infection groups. Patients in RV/EV + multiple viral co-infection group showed a shorter length of PICU stay compared to the other two groups. The article is logically organized with demographic, clinical, microbiological characteristics, and univariate analysis. However, the overall result is not particularly novel and the sample size is quite small.
- Line 132-135, The statement that " In contrast, the 8 patients with RSV and multiple viral co-infections had a significant shorter PICU stay (median 8 days (IQR: 5-12) vs 3 (2-4), p<0.01) and a significantly lower rate of them underwent IMV (0/8 vs 8/22, p=0.04) in comparison to those with RV/EV+RSV as the unique viral co-infection" is confusing. The numbers of patients, IQR, and the rate of children who underwent IMV do not match that in Table 1. Are those typos?
- The authors report that multiple viral co-infection was associated with less clinical severity. However, the median age of the multiple viral co-infection group was 7.9 months old (IQR: 2.2-22.7 months) which is much older than patients with RV/EV infection and those with RV/EV + RSV co-infection. Is age difference a risk factor here that determines the outcome of P-LOS? If so, please emphasize that in the discussion section.
Author Response
- The statement that " In contrast, the 8 patients with RSV and multiple viral co-infections had a significant shorter PICU stay (median 8 days (IQR: 5-12) vs 3 (2-4), p<0.01) and a significantly lower rate of them underwent IMV (0/8 vs 8/22, p=0.04) in comparison to those with RV/EV+RSV as the unique viral co-infection" is confusing. The numbers of patients, IQR, and the rate of children who underwent IMV do not match that in Table 1. Are those typos?
The eight patients reported in that paragraph belong to a little subgroup of the RV/EV + multiple viral co-infection group (18/71 of the total sample, and in 8/18 RSV was also detected). For that reason, as a little subgroup did not appear in Table 1. However, we think it is important to make this differentiation in the text because these 8 patients had different clinical outcomes in comparison to those with RV/EV+RSV as the unique viral co-infection.
According to this comment and other reviewer’s suggestion a new table (New Table 2) has been added to clarify it.
- The authors report that multiple viral co-infection was associated with less clinical severity. However, the median age of the multiple viral co-infection group was 7.9 months old (IQR: 2.2-22.7 months) which is much older than patients with RV/EV infection and those with RV/EV + RSV co-infection. Is age difference a risk factor here that determines the outcome of P-LOS? If so, please emphasize that in the discussion section.
PICU length-of-stay (P-LOS) was defined in the present study above the median of the overall sample (5 days). In Table 2 (new Table 3) results regarding outcome variables were reported. There were no differences in median age between groups of P-LOS (P-LOS <5days 2.1 months, IQR 0.9-14.8; P-LOS >5 days 2.2 months, IQR 1.2-5.6, p-value= 0.836). For that reason, this variable was not considered for the multivariable analysis, so we do not think age difference between groups of viral co-infection could determine P-LOS.
We think that the present version of manuscript includes the answers to the main questions of the reviewers and we hope that it could be accepted for publication in Viruses.
I will be looking forward to hearing from you.
Sincerely yours,
Cristian Launes, MD, MSc, PhD
Round 2
Reviewer 1 Report
Comment: In relation to the discussion lines 255-280
I think the authors should also consider a simpler reason of why they are observing these patterns. Specifically, the authors have observed that:
- Children with multiple virus codetection were significantly older [7.9(2.2-22.7)].
- They had more often previous episodes of LRTI, wheezing, and fever
- They presented symptoms for a longer time period before PICU admission.
- They had a shorter PICU stay with a lower rate of IMV and shorter P-LOS
Obviously, these children had more symptomatic occurrences with different respiratory pathogens up to the point of inclusion in the study. This fact in addition to their higher age point towards a better trained innate immune system at the epigenetic, genetic, and transcriptional level but also a heightened protection from subsequent challenges by the same pathogen or group of related pathogens through the adaptive immunity. Even previous activation of immunity when it happens in proximity to a subsequent infection event can modulate the outcome of the infection. There is also the chance that PCR with its great sensitivity could be picking up traces of degrading genomes from previous pathogen occurrence events if they happened relatively close to baseline.
As a result of the above, these kids are simply doing better than the children with less pathogen occurrences throughout their life time. So, when it comes to RV/EV with or without RSV, which I am convinced that they are the majority of infection-causative agents, they infect either children with a more naïve immune system and/or children which have a better arsenal to combat the infection.
To this end the authors themselves state that “Pathogen-positive children are increasingly symptomatic with decreasing age” in line 291.
It would be informative for the readers to include a paragraph in the discussion about this argument before going into the virome/microbiome interaction with acute viral infections.
Author Response
Reviewer 1
Comment: In relation to the discussion lines 255-280
I think the authors should also consider a simpler reason of why they are observing these patterns. Specifically, the authors have observed that:
Children with multiple virus codetection were significantly older [7.9(2.2-22.7)].
They had more often previous episodes of LRTI, wheezing, and fever
They presented symptoms for a longer time period before PICU admission.
They had a shorter PICU stay with a lower rate of IMV and shorter P-LOS
Obviously, these children had more symptomatic occurrences with different respiratory pathogens up to the point of inclusion in the study. This fact in addition to their higher age point towards a better trained innate immune system at the epigenetic, genetic, and transcriptional level but also a heightened protection from subsequent challenges by the same pathogen or group of related pathogens through the adaptive immunity. Even previous activation of immunity when it happens in proximity to a subsequent infection event can modulate the outcome of the infection. There is also the chance that PCR with its great sensitivity could be picking up traces of degrading genomes from previous pathogen occurrence events if they happened relatively close to baseline.
As a result of the above, these kids are simply doing better than the children with less pathogen occurrences throughout their life time. So, when it comes to RV/EV with or without RSV, which I am convinced that they are the majority of infection-causative agents, they infect either children with a more naïve immune system and/or children which have a better arsenal to combat the infection.
To this end the authors themselves state that “Pathogen-positive children are increasingly symptomatic with decreasing age” in line 291.
It would be informative for the readers to include a paragraph in the discussion about this argument before going into the virome/microbiome interaction with acute viral infections.
Reply
We agree with the reviewer that these observations are of interest to the reader. Nonetheless, we observed that older age and having previous episodes of LRTI were not protecting those children in the univariable analysis (Table 3). The multiple viral detections were the only variable related to this less severity, according to the statistical analysis. However, we agree that some of these detections could be reflecting recently passed infections.
Following these suggestions, we have added new text and two more references (a recent nice review about the known and unknown aspects of immune training, and one big study about repeated viral infections and severity published by the group of Dr. Janet A. Englund, one of the most internationally recognized investigators of children’s viral respiratory infections):
Page 9, line 230: “The immunological, metabolic, and epigenetic processes that mediate trained immunity are still widely unknown [11] but this decreased severity could be reflecting a better trained immune system in older children with previous episodes of LRTI. Despite older age and having previous episodes of wheezing were not protecting children from severe outcomes in this study, detection of these multiple other viruses could correspond to traces of degrading genomes from previous events. Yet the evidence that successive viral infections are milder in children with repeated episodes of LRTI is unclear [12].”
11 - Nat Rev Immunol 2020; 20: 375 – 388. https://doi.org/10.1038/s41577-020-0285-6
12 - J Pediatric Infect Dis Soc 2020; 9: 21 – 29. https://doi.org/10.1093/jpids/piy107
Thank you again for all your suggestions and helping us to improve this manuscript.
Sincerely yours,
Cristian Launes, MD, MSc, PhD
Reviewer 2 Report
Specific comments are as follows:
This article speculates the RV/EV alone detection was observed in young children with severe disease in young children. While, multiple viral co-infection may result in less reduced clinical severity. The subject of this paper is of widespread interest to the scientific community. There have recently been a large number of papers published on viral multiple detections that are commonly observed in paediatric patients with LRTI, but there is no consensus on the relationship between viral co-infections and disease severity. This article fills up the gap in this section,which has a certain innovation. After the revision, the english level of this manuscript has been improved. Therefore, it can be accepted after minor revision.
Author Response
Thank you for your comments.